
# Impact of downscaled rainfall biases on projected runoff changes

Stephen P. Charles[1], Francis H.S. Chiew[2], Nicholas J. Potter[2], Hongxing Zheng[2], Guobin Fu[1], Lu Zhang[2]

[1] CSIRO Land and Water, Floreat WA, 6148, Australia
[2] CSIRO Land and Water, Canberra ACT, 2601, Australia

*Correspondence to*: Stephen P. Charles (steve.charles@csiro.au)

**Abstract.** Realistic projections of changes to daily rainfall frequency and magnitude, at catchment scales, are required to assess the potential impacts of climate change on regional water supply. We show that quantile-quantile matched (QQM)
bias-corrected daily rainfall from dynamically downscaled WRF simulations of current climate produce biased hydrological simulations, in a case study for the State of Victoria, Australia (237,629 km$^2$). While the QQM bias correction can remove bias in daily rainfall distributions at each 10 km$^2$ grid point across Victoria, the GR4J rainfall-runoff model underestimates runoff when driven with QQM bias-corrected daily rainfall. We compare simulated runoff differences using bias-corrected and empirically scaled rainfall for several key water supply catchments across Victoria and discuss the implications for
confidence in the magnitude of projected changes for mid-century. Our results highlight the imperative for methods that can correct for temporal and spatial biases in dynamically downscaled daily rainfall if they are to be suitable for hydrological projection.

## 1 Introduction

Modelled hydrological response to climate change, conditional on regional climate projections, can inform water-supply
planning for resilience on multi-decadal and longer timescales. Global climate models (GCMs) provide broad scale projections (length scales of 100+ km) that are too coarse for direct use in hydrological modelling, hence GCMs are often dynamically downscaled using regional climate models (RCMs) to provide regional (~10 km) scale projections. RCMs can better capture the spatial variability in rainfall change at the scale of catchment response, particularly where there is high spatial variability in rainfall due to orography (Ekström et al., 2015;Grose et al., 2015;Casanueva et al., 2016;Di Luca et al.,
2016;Rummukainen, 2016). However, RCM rainfall characteristics such as daily distributions and sequencing often do not match observations sufficiently well for them to be used directly as input to hydrological models (Maraun et al., 2010;Ehret et al., 2012;Rasmussen et al., 2012;Muerth et al., 2013;Räty et al., 2014). Thus 'bias correction' methods are commonly





applied to adjust RCM rainfall output to match certain characteristics of the observed rainfall (Teutschbein and Seibert, 2012). There is no 'perfect' bias correction approach, thus subjective decisions on their application together with methodological limitations are a source of uncertainty in bias corrected RCM rainfall and the resulting hydrological changes simulated using them (Lafon et al., 2013;Teutschbein and Seibert, 2013;Teng et al., 2015;Ivanov and Kotlarski,

2017;Maraun and Widmann, 2018;Potter et al., 2018). Reported limitations of many daily rainfall bias correction methods include the inability to correct for biases in multi-day rainfall totals, or the daily sequencing of wet and dry days (Chen et al., 2013;Addor and Seibert, 2014;Evans et al., 2017). More generally, bias correction cannot correct for RCM errors inherited from GCM errors in seasonality, temporal sequencing, or large-scale circulation biases that could result in unphysical climate projections (IPCC, 2015).

There is also debate and uncertainty due to the modifications in the climate change signal (CCS) caused by the bias correction (Ivanov and Kotlarski, 2017;Hagemann et al., 2011;Dosio et al., 2012;Themeßl et al., 2012;Velázquez et al., 2015;Mbaye et al., 2016;Switanek et al., 2017;Ivanov et al., 2018;Sangelantoni et al., 2018). Teng et al. (2015) determined that bias correction altered the change signal for many characteristics, including high rainfall amounts, which had a significant impact on simulated runoff and particularly high flows. Ivanov et al. (2018) examined this issue and concluded

that the bias-corrected CCS resulting from correcting RCM intensity distribution bias is defensible.

Here we report on the impact that bias correcting daily rainfall from WRF (Weather Research and Forecasting regional climate model) has on the characteristics of projected future runoff (i) across the state of Victoria in south-east Australia, and (ii) in more detail for ten catchments within Victoria. Potter et al. (2019) found that the quantile mapping (QQM) bias correction approach used to correct raw WRF daily rainfall, applied on a cell by cell seasonal basis, does not correct for

underestimation biases in wet-wet transition probabilities. Hence, we show, BC-WRF rainfall will tend to underestimate runoff compared to runoff simulated using observed rainfall. At the catchment scale both raw and BC WRF rainfall underestimate spatial correlation between cells within a catchment, which is an additional source of runoff uncertainty. Similar results and limitations have been observed for various BC methods in the literature (Lafon et al., 2013;Teng et al., 2015;Maraun, 2016;Rajczak et al., 2016;Maraun et al., 2017).

Thus we investigate the historical performance and CCS of runoff simulated from:



a)  Raw WRF rainfall.

b)  BC WRF rainfall (BC, by season, using QQM).

c)  Observed rainfall that was empirically scaled according to annual raw WRF rainfall changes. For observed rainfall
    we use AWAP (Australian Water Availability Project) 5 km gridded daily climate data, available Australia wide,
interpolated from station data (Jones et al., 2009). The 5 km AWAP daily rainfall was interpolated to the 10 km
    WFR grid and then empirically scaled by the WRF change annual factor, to give future rainfall (as in Chiew et al.
    (2009)).

d)  Observed (AWAP) rainfall that was empirically scaled according to seasonal raw WRF rainfall changes, and then
    rescaled to match annual raw WRF rainfall changes.

e)  The seasonal BC WRF rainfall from (b) twice rescaled to firstly match raw WRF seasonal rainfall changes and then
    additionally rescaled to match raw WRF annual rainfall changes.

As (a), (c), (d) and (e) have the same annual rainfall changes, this allows us to assess the impact on runoff projections of the
choices of using BC versus empirical scaling. We show how the mean, high- and low-flow change characteristics are
influenced by these choices.

## 2 Study area, data and methods

The state of Victoria in south-eastern Australia (Figure 1) experiences large interannual to decadal climate variability that
has a significant impact on water availability for agriculture and water supply for towns and cities (Kiem and Verdon-Kidd,
2010;Chiew et al., 2014). Given these sensitivities to climate variability and change, a research partnership (Victorian
Climate Initiative, VicCI) between Victoria's State Government (Department of Environment, Land, Water and Planning,
DELWP) and Australian research organisations (Bureau of Meteorology and CSIRO) was initiated to ensure that water
policies and management decisions were informed by the most up-to-date earth systems and climate change science (Hope et
al., 2017). VicCI produced projections using daily empirical scaling (Potter et al., 2016). In this study we investigate QQM-
BC dynamically downscaled WRF rainfall simulations for their ability to reproduce simulated runoff for observed conditions
across the whole state of Victoria and also, in more detail, for ten water supply catchments selected on the basis of their
hydrological model performance in reproducing observed runoff (study area and catchments shown in Figure 1).



<Figure 1 here>

## 2.1 Downscaled inputs to hydrological modelling

We use downscaled simulations from the NARCliM project (NSW/ACT Regional Climate Modelling project http://www.ccrc.unsw.edu.au/sites/default/files/NARCliM/index.html). Evans et al. (2014) described the experimental

design for these WRF downscaled simulations, outlining the selection of four GCMs (CCCM3.1, CSIRO-Mk3.0, ECHAM5 and MIROC3.2-medres) used for the boundary conditions for producing future projections and three WRF configurations (R1, R2 and R3) that vary the combinations of planetary boundary layer, surface layer, cumulus and short-wave/long-wave radiation physics used.

Ji et al. (2016) assessed the three WRF configurations, driven by NCEP/NCAR reanalysis boundary conditions, against

AWAP (Jones et al., 2009) observations. They concluded that the R2 simulations performed best in terms of reproducing rainfall seasonal cycles, interannual and decadal variability and spatial patterns over southeast Australia, while noting biases in rainfall amounts can be substantial in some seasons and regions. Olson et al. (2016) also referred to significant WRF biases in rainfall climatology, noting that this was not surprising given WRF configuration selection was on the basis of skill for selected storm events of two week periods, rather than performance at the climatological scale.

Given these findings, we have used BC WRF R2 simulations over Victoria for reanalysis (NCEP/NCAR and ERA-Interim for 1990-2009) and GCM historical (for 1990-2009) and future SRES A2 (for 2060-2079) forced simulations. Note we have only assessed the impact of WRF rainfall bias correction and changes, using AWAP-derived monthly potential evapotranspiration in all cases, to allow us to examine the impact of rainfall properties and changes in isolation from other confounding factors such as temperature and PET change. We have designated the rainfall simulated by WRF model as

'raw' rainfall, to differentiate from the bias-corrected rainfall (henceforth designated 'BC-rainfall'). The resulting runoff, simulated using the raw or BC rainfall inputs, are thus designated 'raw-runoff' or 'BC-runoff', noting 'BC' refers to the use of bias-corrected rainfall inputs to rainfall-runoff modelling.

## 2.2 Hydrological model

We undertook the hydrological modelling experiments using the GR4J rainfall-runoff model (Perrin et al., 2003). The GR4J

model is based on unit hydrograph principle and has been found to be competent in hydrological simulation for a large


number of catchments globally. It has four parameters representing maximum capacity of the soil moisture storage ($x_1$), interbasin water exchange rate ($x_2$), maximum routing storage ($x_3$) and time base of unit hydrographs ($x_4$).

In this study, for rainfall-runoff simulation at each grid cell, the GR4J model was first calibrated against observed daily

streamflow at 137 unimpaired catchments in the region for the period 1981-2010. The calibrated parameters are then applied for each grid cell in Victoria using the nearest neighbour parameter sets (Chiew et al., 2017;Chiew et al., 2018). The objective function of model calibration is defined as (Viney et al., 2009):

$$NSE\_Daily\_Bias = (1 - NSE) + 5[ln(1 + Bias)]^{2.5} \qquad (1)$$

where,

$$NSE = 1 - \frac{\sum_{i=1}^{n}(Q_{mod,i} - Q_{obs,i})^2}{\sum_{i=1}^{n}(Q_{obs,i} - \bar{Q}_{obs})^2} \qquad (2)$$

$$Bias = \frac{(\bar{Q}_{mod} - \bar{Q}_{obs})}{\bar{Q}_{obs}} \qquad (3)$$

where, $Q_{mod}$ is modelled daily streamflow, $Q_{obs}$ is observed daily streamflow, $\bar{Q}_{mod}$ is mean modelled streamflow, $\bar{Q}_{obs}$ is mean observed streamflow, and *n* is total number of days in the modelling period.

Additionally, for a catchment-scale investigation regarding spatial correlation of rainfall within the catchment, two methods for rainfall-runoff model calibration are compared, a 'distributed' method and a 'lumped' method. For the distributed

method, simulations are run for each 10 km grid cell within each catchment using BC-rainfall as input and the runoff simulated for the cells are averaged (with proportional weighting for cells partially within the catchment) to produce catchment-mean runoff, which is then assessed against the AWAP-simulated daily streamflow (mm/day) at the outlet gauge of the catchment. For the lumped method, GR4J is calibrated using a single areal mean daily rainfall time series obtained by averaging AWAP rainfall for the grid cells within the catchment (with proportional weighting for cells partially within the

catchment). Corresponding BC-lumped rainfall from WRF is used as the input to GR4J to simulate runoff for each catchment. Table 1 shows that calibration results of the two methods are comparable at most of the ten catchments. It is interesting to note that the mean annual runoff from the distributed modelling is generally slightly greater than the runoff from the lumped modelling. The lumped modelling generally produced a slightly better calibration than the distributed modelling (Andréassian et al., 2004).

<Table 1 here>

Hereafter, we use the term 'observed' runoff simulations where AWAP rainfall and PET for the 1990-2009 period were inputs to the GR4J model. The observed runoff simulations were considered the benchmark to assess the simulated runoff using dynamically downscaled WRF BC-rainfall for reanalysis-forced and GCM-forced historical runs for the 1990-2009 period. WRF BC-rainfall for GCM-forced 2060-2079 runs were used to produce simulated projections of future runoff.

**3 Results**

**3.1 Historical simulations**

The raw WRF rainfall was found to have large absolute biases compared to AWAP rainfall (Figure 2a) and therefore deemed unsuitable for direct use in hydrological modelling. Bias correction (BC), using daily quantile-quantile-matching (QQM) applied on a seasonal basis (Potter et al., 2019), greatly improved WRF rainfall in terms of annual (Figure 2b) and seasonal

means.

After bias correction, WRF-BC rainfall still had a slight overestimation of annual rainfall relative to AWAP, of around 3 mm averaged across Victoria, which is smaller by up to two orders of magnitude compared to the raw WRF rainfall bias. There was also some spatial consistency to this bias, with both reanalysis- and GCM-forced WRF-BC rainfall showing highest overestimation bias (of up to 50 mm) in the south, east of 146° E, adjacent to an area of underestimation (of up to -15 mm)

immediately to the east, i.e. towards the south-east of the State (Figure 2b). This residual bias is caused by approximation in the interpolation for the highest quantiles (e.g. 99[th] percentile and above), as discussed in Potter et al. (2019).

<Figure 2 here>

Additionally, Potter et al. (2019) has shown that the QQM-BC approach does not correct for other particular rainfall characteristics. These include the underestimation of (i) wet-wet day transition probabilities (i.e. frequency of consecutive

wet days), (ii) multi-day rainfall accumulations and (iii) spatial correlation of rainfall events. Thus, despite WRF-BC slightly overestimating annual mean rainfall compared to AWAP (Figure 2b), the underestimation of wet-wet day transition probabilities and multi-day rainfall accumulations resulted in an underestimation of simulated runoff when compared to runoff simulated using observed (i.e. AWAP) rainfall inputs, as shown in Figure 3. There was an underestimation of mean annual runoff, high flows (99[th] percentile daily flow) and the number of high flow days (days above 95[th] percentile flow) in



most cases (Figure 3). The magnitude of the underestimations was larger for the GCM-forced runs compared to the

reanalysis-forced runs, and (in contrast) NNR-based results for the south-east produced some overestimation biases. The

magnitudes of these biases are compared to the magnitudes of their respective climate change signals (projected future minus

historical) in the next section.

<Figure 3 here>

### 3.2 Projected future simulations

The influence that the bias correction (QQM-BC) has had on WRF rainfall change was assessed by comparing the WRF-BC

change (BC future – BC historical) to that obtained from empirically-scaled (ES) change (i.e. from applying the raw future –

raw historical change to AWAP historical). The ES AWAP rainfall, scaled separately according to both raw WRF (i) annual

and (ii) seasonal rainfall changes, produced 'annual ES change' and 'seasonal ES change', respectively.

Differences between the annual rainfall changes obtained from the annual ES (Figure 4a) and seasonal ES (Figure 4b)

indicated differences between WRF and AWAP rainfall seasonality. For example, in the case of the ECHAM5-forced results

there was a Victoria-average annual rainfall increase of 4.8 mm using seasonal ES compared to 9.6 mm using annual ES.

Thus ECHAM5-forced WRF must have produced too much rain in one or more seasons, compared to AWAP seasonality, to

result in this discrepancy between seasonal and annual ES changes. That is, if the proportion contributed by each season had

been similar between ECHAM5-forced WRF and AWAP, then the seasonal and annual ES would produce similar annual

changes, as can be seen for the remaining three GCM-forced cases.

The WRF-BC future rainfall was also re-scaled for comparison with ES results (rescaled twice, seasonally then annually) so

as to match the WRF-raw annual rainfall change signal, as shown in Figure 4c (i.e. the change signal in mean annual rainfall

is the same in Figures 4a, b and c). In general the WRF-BC rainfall changes (Figure 4d) were wetter (or less-dry) changes

than both the WRF-BC re-scaled and the ES changes (Figure 4a,b,c). In the example of ECHAM5-forced results, the

seasonal ES gave a 4.8 mm Victoria-average increase whereas BC gave an increase of 16.7 mm, with the changes showing

similar spatial patterns but with BC giving larger increases particularly over the high altitude areas.

It was also possible for WRF-BC-rainfall change to have a different change direction to that of the raw WRF rainfall, as

evident for MIROC3.2-forced results in the north-east Victorian Alps where the BC has a positive (i.e. wetter) change in





contrast to the ES (i.e. raw WRF) having a negative (i.e. drier) change. Hence there was a difference between the seasonal

ES average decline of -14.9 mm and the corresponding BC increase of 8.6 mm. In contrast, the magnitudes and spatial

patterns of BC and ES changes were much more similar to each other for the results from WRF CCCMA3.1 and CSIRO-

Mk3.0-forced results (Figure 4).

<Figure 4 here>

The impact of these BC and ES rainfall differences on their corresponding simulated runoff changes were assessed by

comparing runoff simulations using four rainfall input variations, namely (a) ES-ann: AWAP rainfall scaled according to

WRF-raw annual rainfall changes; (b) ES-seaann: AWAP rainfall scaled twice, firstly according to WRF-raw seasonal

rainfall changes and then rescaled to match WRF-raw annual rainfall changes; (c) BC-RS: the WRF-BC future rainfall

scaled twice, firstly so as to match WRF-raw seasonal rainfall changes and then rescaled to match WRF-raw annual rainfall

changes; and (d) BC: the WRF-BC rainfall. For (a) and (b) the runoff changes were the difference in simulated runoff using

ES-future relative to AWAP-historical rainfall, whereas for (c) and (d) the runoff changes were the difference in simulated

runoff using BC(-RS)-future relative to BC(-RS)-historical rainfall.

Consistent with the rainfall change differences shown in Figure 4, the mean annual runoff changes (Figure 5) highlight that

(i) BC produced greater runoff increases than ES for locations with increased runoff and, correspondingly, (ii) BC produced

smaller runoff decreases than ES for locations with decreased runoff, and (iii) BC-RS runoff change was drier than BC,

however not as dry as the ES runoff changes. This is due to the combined effect of the remaining biases in BC-RS rainfall

(e.g. underestimation of wet-wet day transitions and multi-day accumulations) and also rainfall characteristics that BC can

change that are not accounted for by ES, such as changes to upper tails of daily rainfall distributions producing more intense

extreme rainfall and sequencing of wet and dry days. That is, for wetter projections BC rainfall can have more frequent wet

days and more intense rainfall extremes relative to ES, leading to larger BC runoff increases compared to ES (e.g.

ECHAM5). The ability of BC rainfall to include changes in temporal characteristics is an important improvement over ES,

given ES is constrained to reproducing the historical sequencing. Thus, for drier projections, such sequencing and upper-tail

changes can mitigate the runoff decreases seen in the ES results (e.g. CCMA3.1 and CSIRO-Mk3.0). In some cases (e.g.

MIROC3.2 for some areas) these factors have changed ES runoff decreases to BC runoff increases (Figure 5). We address our confidence in such changes in the discussion section.

<Figure 5 here>

Correspondingly, the changes to 99th percentile daily runoff (Figure 6) show BC produced larger increases (or smaller

decreases) than ES, and BC has changed the direction from decreases to increases for certain areas for all cases except ECHAM5-forced (which did not produce decreases). The BC-RS changes are similar (slightly less) than BC changes, indicating they are due to BC derived changes in rainfall characteristics, such as changes to upper tails of rainfall distributions, and potentially changes in wet-day sequencing too, both of which influence high flows and were not accounted for by ES.

<Figure 6 here>

Changes to the frequency of high flow days, i.e. number of days greater than the historical 95th percentile daily flow show larger increases and smaller decreases for BC compared to ES results (Figure 7). BC-RS are changed compared to the BC results, however in most cases they remain closer to the original BC than ES results. One exception is MIROC3.2-forced results, where large BC increases in the NW are absent in BC-RS.

<Figure 7 here>

### 3.3 Catchment scale simulations

The change signal magnitudes are highly dependent on the driving GCM, with noticeable differences between lumped and distributed cases. Figure 8 shows a general pattern that when there is a projected increase in runoff, in all cases the lumped results give larger increases than the distributed case. Likewise for projected decreases, there are greater decreases (i.e. more

negative) for the lumped than for the distributed cases.

ECHAM5-forced results consistently simulates large runoff increases of at least +10% and up to a +32% increase. CSIRO-Mk3.0 consistently simulates the largest runoff decreases, of up to -29%, and CCCMA3.1 simulates decreases (of up -32%) for all catchments except for 403210. MIROC3.2 WRF-BC-rainfall produces smaller changes, with a range from -12% to +10%.

<Figure 8 here>

Given the historical runoff underestimation biases shown earlier (Figure 3), we look at the bias in the simulation of high flows for the ten catchments, for the distributed and lumped calibrations (Figure 9). There are small differences for most of the percentiles shown, with a small underestimation of the $90^{th}$ percentile becoming greater and more variable for the $99^{th}$ percentile in all cases. The distributed $99^{th}$ percentiles have slightly more underestimation than those for the lumped for the

reanalysis run, whereas for the four driving GCMs results the lumped results show slightly more underestimation than the distributed. This underestimation of high flows ($90^{th}$ percentile and above) seen in most cases will result in underestimation of annual and monthly runoffs, with the reasons for this underestimation discussed later. The corresponding projected changes, shown in Figure 10, show small decreases for CCCMA3.1 up to the $90^{th}$ percentile and a large range from decreases to increases for the $95^{th}$ and $99^{th}$ percentiles. CSIRO Mk3.0 presents a more consistent decrease with higher

percentiles, with a larger range for the $99^{th}$ with at least one catchment experiencing an increase. ECHAM5-forced results presents the most consistent projected changes, with increases particularly for the $99^{th}$ percentile. Relatively small changes, with mainly decreases for higher percentiles, are seen for MIROC3.2-forced results.

<Figure 9 here>

<Figure 10 here>

**4 Discussion**

Potter et al. (2019) have shown that WRF-BC rainfall, while greatly improved over WRF-raw rainfall (Figure 2), underestimates sequences of wet days, multi-day accumulations and daily rainfall spatial correlation. We show that these remaining biases result in the underestimation of simulated historical mean seasonal and annual runoff and high flows (Figure 3). Whether these biases influence the magnitude of projected runoff change is the focus of the discussion presented

here. Given that the characteristics of the WRF-raw rainfall changes are modified by QQM-BC (Figure 4) and hence the WRF-BC rainfall derived runoff changes are different (Figure 5), there is a need to assess whether they are more or less realistic than the runoff changes produced simply by empirically scaling the historical rainfall series by the annual or seasonal change signal in the WRF raw rainfall. Note that such a comparison does not rely on the ES derived runoff changes being correct, i.e. we are not validating the BC derived runoff changes against those from ES. We are merely attempting to

determine whether BC runoff changes are more or less credible, in terms of the characteristics assessed herein.





There is an argument that, because the QQM-BC rainfall corrects distributional biases in the raw WRF rainfall, the QQM-BC rainfall change signals are more realistic than those of the original WRF-raw rainfall. Hagemann et al. (2011), investigating hydrological changes globally, has noted that for some regions the magnitude of change in climate change signal due to bias correction can be greater than the magnitude of the signal itself, such that bias correction uncertainty can

be as large as climate model uncertainty. Several subsequent studies have investigated how bias correction modifies the rainfall climate change signal (Dosio et al., 2012;Ivanov et al., 2018;Mbaye et al., 2016;Potter et al., 2018;Sangelantoni et al., 2018;Themeßl et al., 2012). Themeßl et al. (2012) concluded that QQM-BC is likely to improve the reliability of projected changes if the climate model biases are related to the shape of the distribution i.e. when RCM bias is magnitude-dependent. Dosio et al. (2012), investigating ENSEMBLES RCMs over Europe, noted that the RCMs had a tendency to

overestimate extreme rainfall and hence the increases in P99 for their bias-corrected results were two to three times smaller than the original RCM's increases. Mbaye et al. (2016) also found that BC reduced the changes in heavy rainfall events. Ivanov et al. (2018) concluded that changes to the CCS due to bias-correction are scientifically appropriate and therefore the BC CCS is more trustworthy than the raw CCS, due to the removal of model biases that adversely influence the original CCS. Given the large biases shown in Figure 2, we did not have confidence in using WRF raw rainfall as input to

hydrological modelling, thus our findings implicitly agree with these previously published conclusions.

Regarding the BC simulated hydrological changes for Victoria, comparison of the magnitude of BC mean runoff bias (in the historical period) (Figure 3, top row) to BC mean runoff change (under projected climate change) (Figure 5, bottom row) shows cases where the absolute value of Victorian area-average change is larger than the bias (CSIRO-Mk3.0-forced change -8.7 mm, bias -6.5 mm; ECHAM5-forced change 15.4 mm, bias -6.2 mm) and cases where the change is smaller than the

bias (CCCMA3.1-forced change -0.2 mm, bias -5.5 mm; MIROC3.2-forced change 5.23 mm, bias -8.2 mm). However the range of bias is much smaller than the range of change, as evident by the scales on the respective plots (bias ranging from -30 to +20 mm; change ranging from -150 to +150 mm). Hence the bias is smaller and spatially consistent compared to the change signal.

The differences in runoff changes from empirically scaled rainfall (i.e. based on seasonal and/or annual changes in WRF-raw

rainfall) and BC rainfall were shown in Figure 5 to Figure 7. For annual mean runoff, daily 99[th] percentile flow and the





number of days above observed 95th percentile flow, respectively. Comparing ES-ann to ES-seaann derived runoff changes shows similar mean runoff change (Figure 5), with three GCM-forced very similar and ECHAM5-forced change larger for ES-seaann (6.2 mm) than for ES-ann (4.1 mm). For Q99 (Figure 6), the area-average values are similar however larger changes (cases of both decreases and increases) are evident for certain regions.

Comparing ES-seaann to BC-RS derived runoff changes shows BC-RS mean runoff change is wetter than ES-seaann in all four cases (Figure 5). The ability of BC-RS to include (through the BC, not the RS) changes to the upper tail of rainfall distributions and wet-day sequencing, in contrast to ES which cannot, is likely to be the reason BC-RS is wetter. Correspondingly, Q99 changes for BC-RS are more positive than for ES-seann (Figure 6).

Comparing BC-RS to BC derived runoff changes shows BC mean runoff change to be wetter than BC-RS, again in all four
cases (Figure 5). This supports the case that ES limitations are causing drier projections than BC, given that BC-RS constrained to match ES-seaann changes is drier than BC. In three out of four cases the Q99 changes from BC are more positive than for those from BC-RS (Figure 6). Given Q99 is underestimated for historical BC (Figure 3, 2nd row), this suggests BC changes are more realistic than BC-RS and hence also more realistic than ES changes in Q99, again this could be because BC can modify the upper tail of rainfall distributions and also wet-day sequencing.

Muerth et al. (2013) note that the more strongly biased a climate simulation is, the larger the effect of bias correction on the change signal of hydrological response. We see an example of this in the results from the MIROC3.2 GCM, which had the largest WRF-raw-rainfall bias (Figure 2) corresponding with the largest difference between ES and BC rainfall and runoff change (Figure 4 and Figure 5, respectively), including change of direction from ES runoff decreases to BC runoff increases in north central and north eastern regions. Confidence in such changes requires caution and relies on assuming the
transferability of the BC from the historical to the future period (Velázquez et al., 2015). This assumption of transferability is a caveat on all BC, but particularly impactful in this example.

The BC rainfall remaining biases add uncertainty to the magnitude of runoff changes, as rainfall that correctly reproduced wet day sequencing and multi-day totals would potentially produce different runoff changes. If BC rainfall did not have the wet to wet day transition and multi-day total underestimation biases, then projected increases could be greater and decreases
lesser. Hence confidence in BC results is diminished because of these remaining rainfall biases. That is, while the BC rainfall

change could be more realistic than WRF-raw and ES changes, for the reasons noted above, they would be more credible without the remaining biases in these rainfall characteristics.

At catchment scales, spatial correlation in WRF-raw and WRF-BC rainfall is underestimated and thus runoff is underestimated. This is an additional uncertainty and unaccounted source of bias in catchment runoff change, also leading to underestimation in projected runoff changes (Figure 8). For historical-period simulations, the annual runoff has a longer and greater upper tail for the lumped simulations compared to the distributed cases. Correspondingly, the lumped upper quartile ranges are greater and the lower quartile ranges are smaller relative to distributed. Seasonally, for the low-flow summer and autumn seasons, the lumped flow has larger range than distributed. For winter, lumped simulations produce greater median flows and upper tails. In spring, median and upper tail differences are less pronounced with smaller lower tails in the lumped simulations. Regarding the climate change signal, annually, lumped simulations tend to produce wetter changes than distributed simulations (an exception - the dry CSIRO-Mk3.0 is slightly drier for lumped). Seasonally, runoff changes are small (in absolute terms) for summer and autumn. Consistent dry projections are seen for spring, with lumped slightly drier than distributed in contrast to other seasons. For winter, lumped is similar to distributed, wetter projections having longer upper tails.

As shown in Figure 9 and Figure 10, for CSIRO Mk3.0-forced results the combination of an underestimation of the highest daily flows for historical conditions and a projection for decreases in these high daily flows in the future means the runoff change may be overestimated (i.e. too large a projected decrease in runoff). For ECHAM5-forced results, which also underestimates the historical high flows, the projected increases may inherently underestimate the runoff increases.

**5 Conclusions**

Using WRF-BC rainfall from historical GCM-forced simulations to drive GR4J models produces underestimates of observed runoff. This underestimation is because the spell-lengths of consecutive wet days are underestimated by WRF-BC rainfall, compared to observed, and hence the upper tail of the runoff distribution is underestimated.

For projected climate change impact on runoff, using raw WRF rainfall would be unrealistic because the runoff overestimates for historical climate mean that any projected increase in rainfall will produce too large an increase in runoff



and any decrease in rainfall will produce too small a decrease in runoff for these particular catchments. For the WRF-BC-rainfall derived runoff changes, where historical runoff is underestimated then an increase in rainfall may underestimate the runoff increase and a decrease in rainfall may overestimate the runoff decrease. This study has attempted to understand and document some of these issues and impacts regarding how RCM BC influences hydrological simulations (Addor and Seibert, 2014), leading to the following conclusions:

1. There is a need for reporting of the caveats and influences that methodological choices have on projected hydrological changes derived from dynamical downscaled rainfall.

2. WRF downscaled rainfall requires bias-correction to be suitable for hydrological model input. QQM-BC can reproduce observed daily rainfall distributions for each grid-cell, however QQM-BC rainfall underestimates wet to wet day transition probabilities, multi-day totals and spatial correlation. These QQM-BC rainfall biases result in runoff biases, with runoff simulations underestimating mean seasonal, annual runoff and high flows.

3. Raw WRF rainfall changes are modified by QQM-BC, and thus runoff changes are modified also. Because the QQM-BC rainfall corrects distributional biases in WRF-raw rainfall, the QQM-BC rainfall change signals are plausibly more realistic than the changes of the raw WRF rainfall.

4. Differences in projected future runoff changes from empirically scaled rainfall (i.e. based on WRF-raw rainfall changes) and QQM-BC rainfall are due to several factors, including limitations in ES not present in BC such as limited ability of ES to change multi-day rainfall distribution upper tails and sequencing. We conclude that BC runoff changes are more realistic than those from ES, with the caveat that the remaining BC rainfall biases due to underestimation of wet sequences, multi-day totals and spatial correlation need to be addressed to provide greater credibility for runoff projection.

5. The QQM-BC rainfall biases influence the magnitude of runoff changes, as discussed above, thus we conclude runoff increases may be underestimated and decreases overestimated. Additionally, as noted at catchment scales, spatial correlation in WRF-raw and hence QQM-BC rainfall is underestimated, an additional source of underestimation of projected runoff changes (Figure 8).



Addor and Seibert (2014) discuss the need to better understand the underlying causes of these biases in climate models as well as a more systematic quantification of their impacts on hydrological response. Other recent studies have also questioned the application of BC without fully understanding the underlying reasons for the biases. Such studies have recommended 'process-based' approaches to evaluate RCM simulation temporal and spatial realism, and thus credibility (Maraun et al., 2017;Maraun and Widmann, 2018). In future work we will assess multiple RCMs for their process performance in this region, with a view to developing rainfall bias correction methods that can reduce biases in hydrological predictions.

## Acknowledgments

This study was funded by the Victorian Department of Environment, Land, Water and Planning (DELWP) and the Australian Commonwealth Scientific Industrial and Research Organisation (CSIRO) through the Victorian Water and Climate Initiative (VicWaCI). We thank Jason Evans, Giovanni Di Virgilio and Fei Ji for providing WRF output and advice.

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



## TABLES

**Table 1 Catchment attributes and distributed and lumped calibration (1990-2009 period) runoff and NSE metrics**

| CATCHMENT ID | AREA (KM²) | MEAN ANNUAL RUNOFF (MM) | | NSE DISTRIBUTED | NSE LUMPED |
|---|---|---|---|---|---|
| | | DISTRIBUTED | LUMPED | | |
| 221212 | 731 | 140 | 145 | 0.785 | 0.783 |
| 226204 | 557 | 251 | 248 | 0.756 | 0.766 |
| 226402 | 608 | 147 | 124 | 0.704 | 0.876 |
| 235203 | 721 | 127 | 126 | 0.690 | 0.697 |
| 235208 | 575 | 271 | 223 | 0.764 | 0.820 |
| 235224 | 1042 | 212 | 210 | 0.798 | 0.866 |
| 403210 | 1229 | 373 | 358 | 0.892 | 0.908 |
| 405209 | 629 | 375 | 385 | 0.853 | 0.868 |
| 405219 | 704 | 400 | 416 | 0.881 | 0.883 |
| 405227 | 627 | 426 | 403 | 0.851 | 0.862 |





**FIGURES**

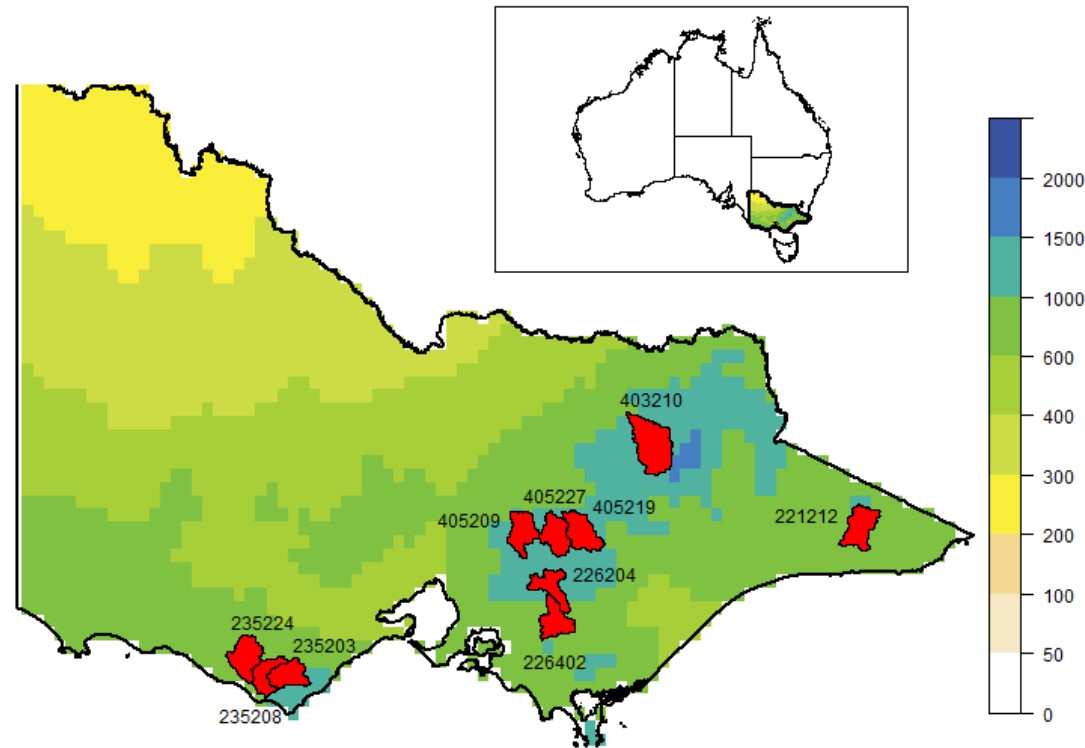

**Figure 1 State of Victoria, Australia, mean annual rainfall (mm/yr for 10 km grid cells from AWAP for 1990-2009) and location of**
5  **study catchments used for rainfall-runoff modelling**

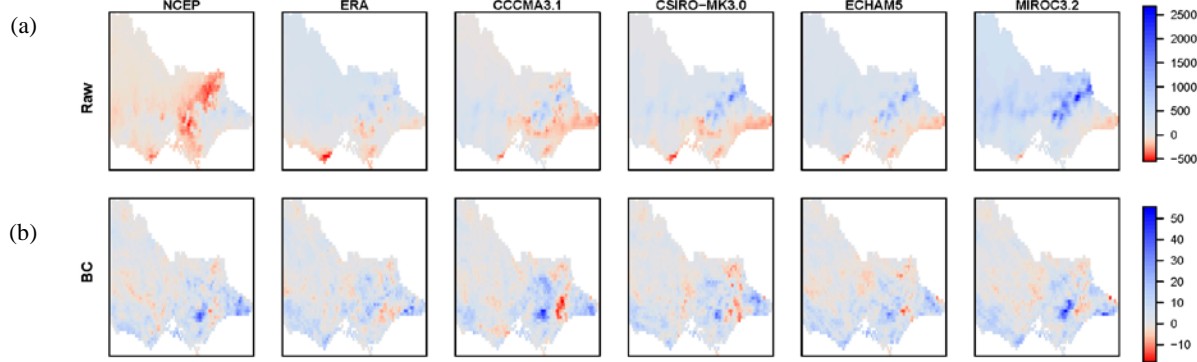

**Figure 2 Historical (1990-2009) mean annual rainfall bias (WRF minus AWAP, mm) for (a) raw WRF and (b) BC WRF. Panels (left to right) are WRF runs forced by two reanalysis (NNR, ERAI) and then four GCMs (CCCMA3.1, CSIRO-MK3.0, ECHAM5 and MIROC3.2)**





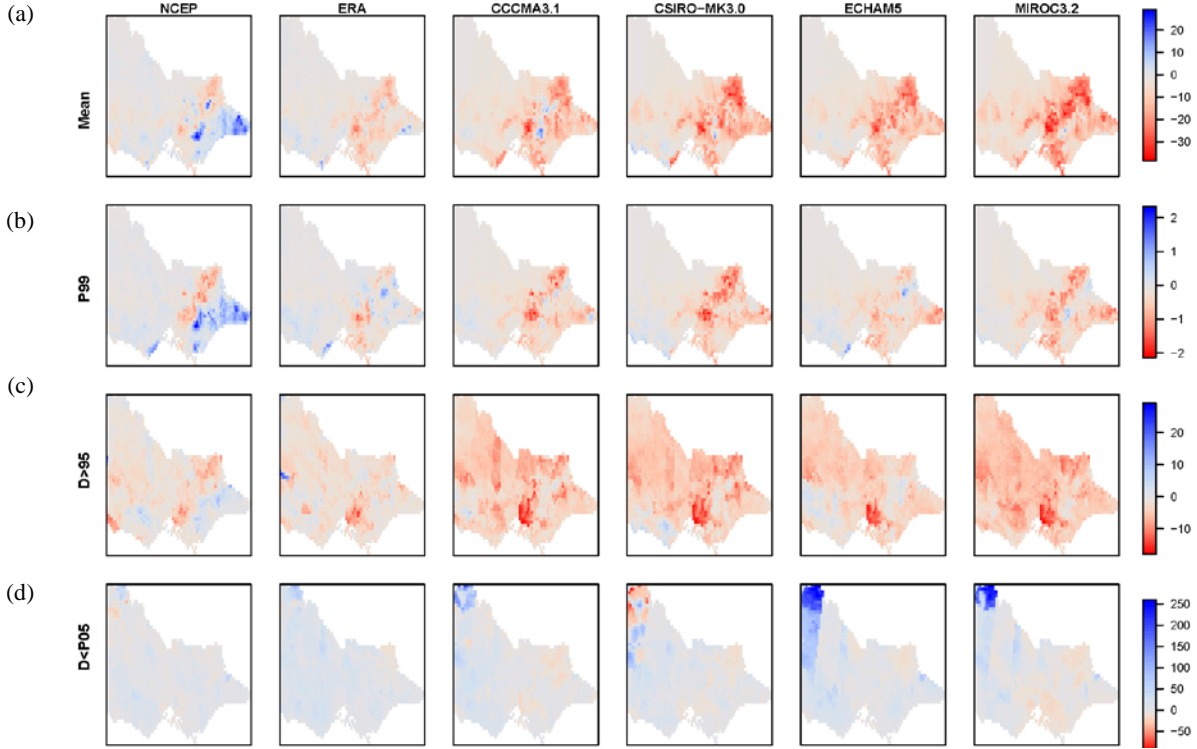

**Figure 3 Historical (1990-2009) runoff simulation bias (i.e. runoff using WRF-BC rainfall minus using AWAP rainfall) for (a) mean annual runoff (mm), (b) daily 99th percentile runoff (mm), (c) number of days exceeding AWAP 95th percentile, (d) number of days below AWAP 10th percentile. Panels (left to right) are using WRF-BC rainfall forced by two reanalysis (NNR, ERAI) and then four GCMs (CCCMA3.1, CSIRO-MK3.0, ECHAM5 and MIROC3.2).**



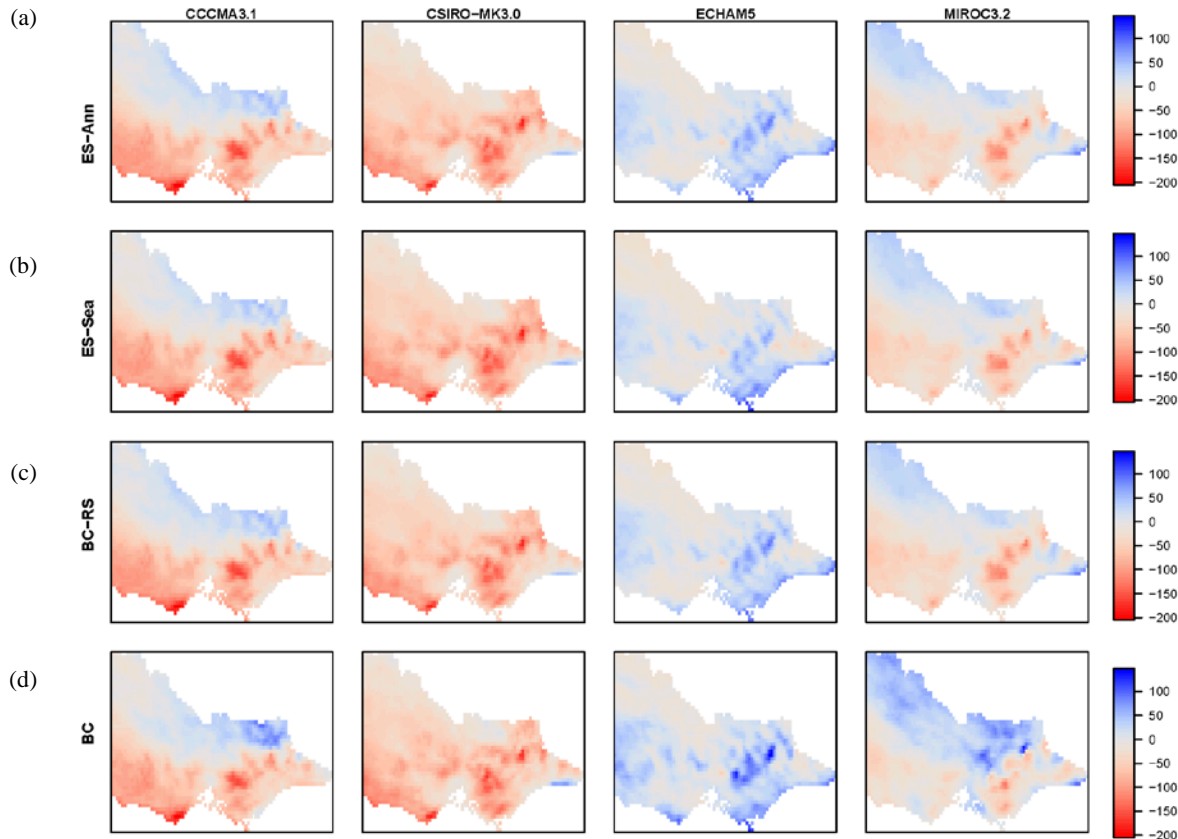

**Figure 4 Mean annual rainfall change (mm) (a) empirical annual scaling, (b) empirical season scaling, (c) BC re-scaled (according to ES seasonal then annual changes), (d) BC. Panels (left to right) CCCMA3.1, CSIRO-MK3.0, ECHAM5 and MIROC3.2**





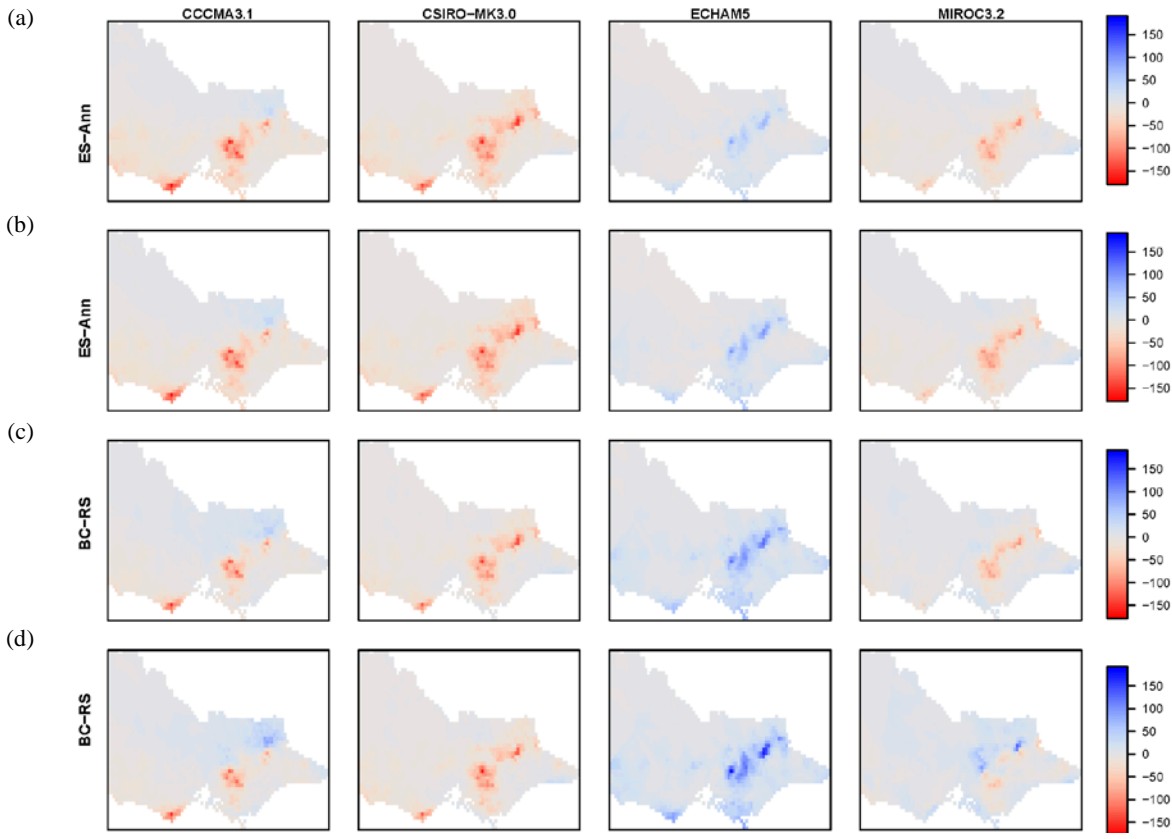

**Figure 5 Mean annual runoff change (mm, Future-Historical) (a) ES-ann, (b) ES-seaann, (c) BC-RS (d) BC. Panels (left to right) CCCMA3.1, CSIRO-MK3.0, ECHAM5 and MIROC3.2.**



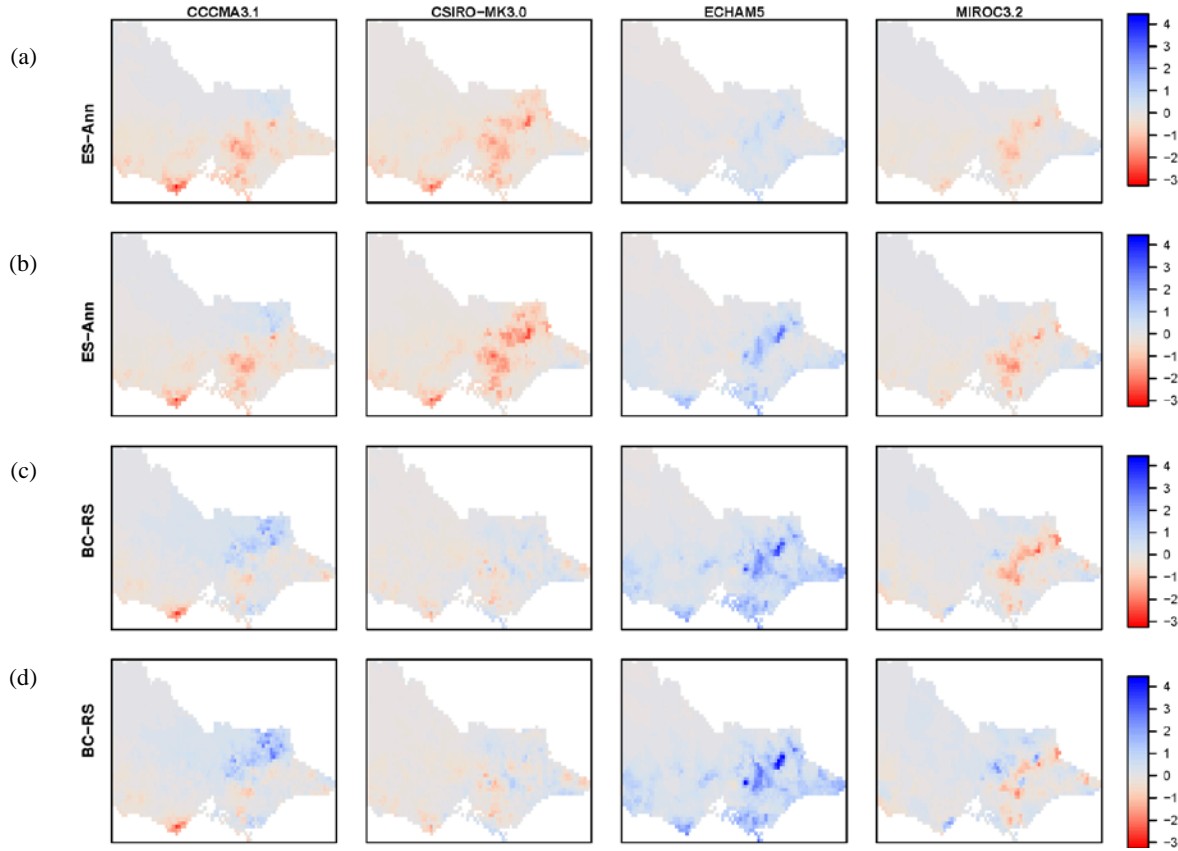

**Figure 6 Change in 99[th] percentile daily flow (mm, Future-Historical) (a) ES-ann, (b) ES-seaann, (c) BC-RS (d) BC. Panels (left to right) CCCMA3.1, CSIRO-MK3.0, ECHAM5 and MIROC3.2.**





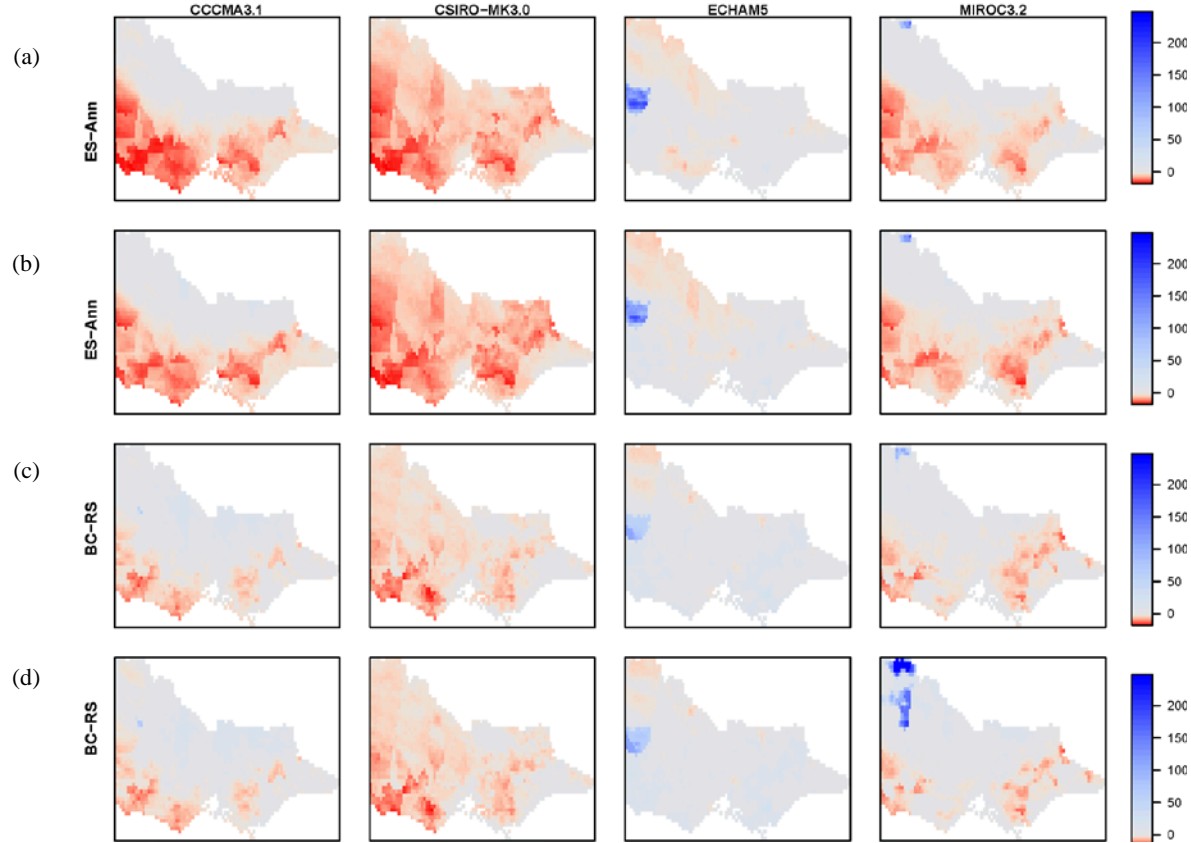

**Figure 7 Change in number of days exceeding historical 95th percentile daily flow (Future-Historical) (a) ES-ann, (b) ES-seaann, (c) BC-RS (d) BC. Panels (left to right) CCCMA3.1, CSIRO-MK3.0, ECHAM5 and MIROC3.2.**





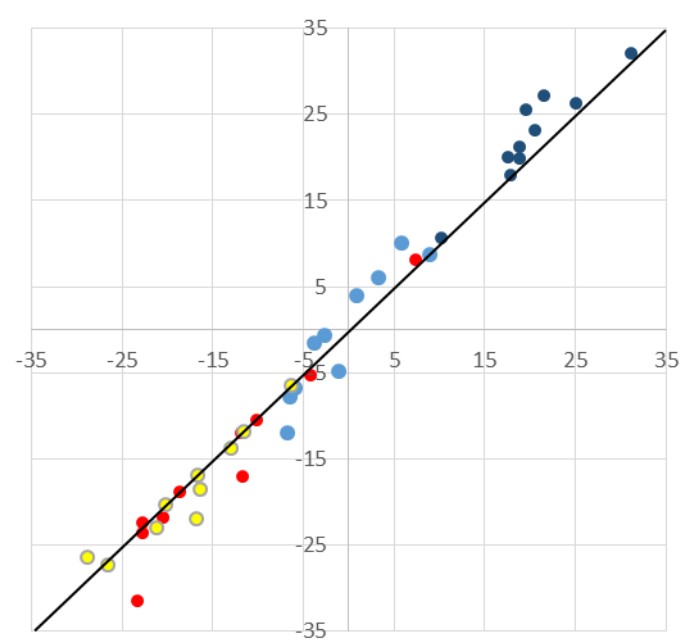

**Figure 8 Distributed (horizontal axis) versus Lumped (vertical axis) change (future period relative to current period) in mean annual runoff (%). Orange = CCCMA3.1; yellow = CSIRO Mk3.0; Dark blue = ECHAM5; Light blue = MIROC3.2**





(a)

(b)

(c)

(d)

(e)

(f)

(g)

(h)

(i)

(j)





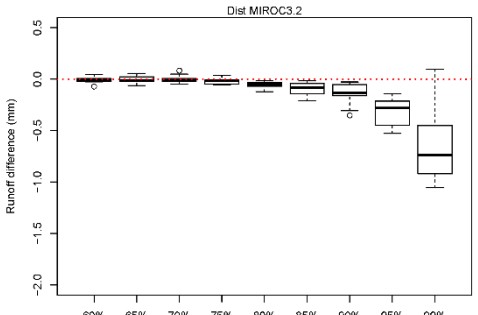
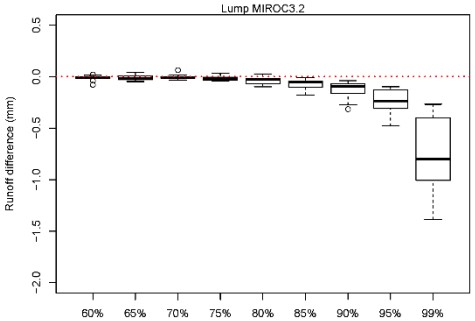

**Figure 9 Bias in historical simulated high flows (WRF-BC-rainfall minus AWAP-rainfall inputs) for distributed (left) and lumped (right) simulations with WRF inputs informed by the four GCMs. Negative values indicate WRF-BC-derived results are smaller than AWAP-derived results. Range of box-plots show results from the ten catchments.**





**Figure 10 Climate change simulated runoff high flow changes (future minus historical using GCM-forced WRF-BC-rainfall inputs). Range of box-plots show results from the 10 catchments.**