# Peer review of "Impact of downscaled rainfall biases on projected runoff changes"

_Hydrology and Earth System Sciences, 2019_

## Short Comment (SC1) · 28 Aug 2019

I appreciate the valuable work done by the authors. The authors have analysed quantile-quantile matching approach of bias correction while using the WRF rainfall data (and characteristic data generated from the WRF data) and assessed their implications on hydrological projections in the Victoria state of Australia. The hydrological model (GR4J) setup was done at both lumped catchment scale and distributed 10 km grid scale. The results are interesting, however, I have a few questions regarding the methods.

General comments: The abstract is nicely written with a valuable conclusion, however, do you think the biases in runoff projection are also limited to the hydrological

model choice? More appropriately, can the biases be reduced by employing a semi- or fully-distributed hydrological models which account for land use, soil characteristics and sub-surface flows? This is because there are several existing studies which have validated that runoff predictions are not only limited to climatological data, but physical processes such as SW-GW interaction, vegetation cover etc. which GR4J does not count for. Based on your experience do you think if the bias correction (while using either QQM or linear scaling) of the GCM projections was done at station scale and used for the catchment scale (lumped) hydrological simulation would have reduced the bias in the results compared to the results presented (as wide range of results are observed for the four GCM projections) in your manuscript? This also follows based on the findings of Muerth et al. (2013) which you have discussed in Pg12 Ln15 as high biased climate signals propagate in hydrological simulations. Pg4 Ln 16: Can you please elaborate on the use of GCM data for only SRES A2 scenario? Also, as the SRES emission scenario data inherit more uncertainties over the RCP scenarios, is it valid for its use in this study as this paper deals with uncertainties cascading to hydro- logical simulations? This is because, in a study such as Woldemeskel et al. (2015), the uncertainty in precipitation for RCP scenarios of CMIP5 were significantly lower com- pared to SRES scenarios of CMIP3 GCMs in the Australian region. What do you think about this? Can you please write more on the calibration of GR4J at grid scale? As per my experience, GR4J accounts for the rainfall and PET to determine the effective precipitation and then the flow routing is done at catchment outlet. As per the descrip- tion given, it is not clear how the model is set using the distributed method at 10 km grid size? My specific question is how the routing is done at 100 km2 grid? Also, how the calibration is done at 100 km2 grid?

Specific comments: Pg2Ln 20 and 21: Please be consistent with the use of "-" in BC WRF as in some places it is there and some places not. Same issue with the "BC- rainfall" and "BC rainfall". Please check. Pg3 Ln6: Typo "WFR" instead of "WRF". Pg3Ln 3-7: I understand there are several advantages of using the AWAP data, how- ever, as per the study of Tozar et al. (2012), using gridded data (be it either AWAP or

SILO) always introduce artificiality (due to spatial and temporal interpolation) and alter the "realness". This further cascades to hydrological simulations and leads to unrealistic results. Do you think using AWAP dataset is valid rather than using BoM gauges data for this study? Tozar C.R., Kiem, A.S., Verdon-Kidd, D.C., 2012. On the uncertainties associated with using gridded rainfall data as a proxy for observed. Hydrology and Earth System Sciences, 16, 1481-1499. Woldemeskel F.M., Sharma A., Sivakumar B., Mehrotra R., 2016. Quantification of precipitation and temperature uncertainties simulated by CMIP3 and CMIP5 models. Journal of Geophysical Research: Atmospheres, 121, 3-17. Can you please briefly write about the quantile-quantile mapping approach used in the study? It will be useful for the readers from non-climatological background. Pg5 Ln4: Can you please provide a list of all these 137 catchments in the supporting information? Can you also please provide the median and the percentile of the GR4J calibration parameters for both catchment scale and grid based calibration in the supporting information file? Pg5 Ln 23-24: Can you briefly mention the probable reasons for the better performance of the lumped approach over the distributed approach of GR4J calibration? Given that the catchments are flat (as majority of catchments in the SE Australian region are flat), a distributed approach should have yielded in better results (Deb 2019; Deb et al. 2019). Deb P., 2019. Modelling non-stationarity in rainfall-runoff relationships in Australian catchments. PhD Thesis, University of Newcastle, Australia. Deb P., Kiem A.S., Willgoose G., 2019. A linked surface water-groundwater modelling approach to more realistically simulate rainfall-runoff non-stationarity in semi-arid regions. Journal of Hydrology, 575, 273-291. Are the runoff simulation results presented in sections 3.1 and 3.2 based on the distributed GR4J model setup? For the future plans (as mentioned in the conclusion), I would also recommend the use of a physical-process based semi-distributed (if not fully distributed) hydrological model for the future hydrological predictions.

---

## Referee Comment (RC1) · Anonymous Referee #1 · 8 Sep 2019

The authors developed a study that analyses the impacts of climate model downscaled rainfall biases on the simulated runoff for the Victoria State in Australia. They use the WRF Regional Climate Model (RCM) driven by two reanalysis datasets and four Global Climate Models (GCMs). They evaluate the raw and bias-corrected (QQM) simulations and scaled observations. In order to evaluate the changes in runoff, they employed the GR4J hydrological model with a lump and a distributed setup.

GENERAL COMMENTS Overall, I can see a heavy load of work involved in this research along with some very interesting results. I have some comments for the authors to consider:

An important part of your work analyses empirically scaled observed data. It might be useful for the readers if you describe a bit of the method from Chiew et al. (2009).

[Figure]

Similar to the above, you could provide a few sentences to describe the QQM bias-correction method.

In future work, such as the one stated at the end of the conclusions, I would recommend using the last generation of climate models that is available as this would provide an analysis of the current state-of-the-art.

I understand that this works builds up from previous studies, however, could you give a sentence on why you (or the studies that your work builds on) choose to use the WRF model? Would you expect similar results from using other RCMs? The above applies for the hydrological model, would you expect that a fully-distributed integrated model will have different result?

SPECIFIC COMMENTS Pg.1 L27 - The comma is missing in the following: Thus, 'bias correction' methods

In Pg. 2 Lines 2 to 9 - You are referring to some of the limitations of bias correction. I think it would be also important to include the stationarity assumption of the relationship between simulations and observations. I think this is the main concept that supports the idea of bias correction.

Pg.4 Lines 16 to 20 - It is not clear to me whether you use the monthly mean potential evapotranspiration or daily values. Also, you could include a sentence stating whether if including the potential evapotranspiration simulations could change your results considerably.

Pg. 6 Line 16 – You could say how large is the bias of the 99th percentile in rainfall as it can give a background for the bias in the 99th percentile of runoff.

Pg 9. Lines 17 to t20 – Can you include some reference that assess whether the model is good to simulate runoff on a changing catchment. I am not sure that there is a reference for the GR4J model. If not, you can acknowledge that there is no previous study analyzing it.

Pg. 11 Lines 1 and 2 – You could also include the opposite view that bias-corrected outputs should be used with care giving that they are not based in any knowledge of the clime at physics (Ehret et al 2012).

Pg. 11 last line and page 12 first line – Correct the sentence as it does not make sense at the moment.

Table 1. Given that the high flows are relevant for this research, consider including here how the models do when simulating the high flows (i.e. 99th percentile). This will increase the credibility of the results (even if you use the simulated, and not the observed, runoff as benchmark for comparison.

Figure 1. You can add the legend title (i.e. annual mean rainfall (mm/yr)"

Figure 9. Add the title of the y axis title.

---

## Referee Comment (RC2) · Eylon Shamir (Referee) · 16 Sep 2019

Reviewer Eylon Shamir. eshamir@hrcwater.org

The manuscript presents a comprehensive comparison of various rainfall input to a hydrologic model in order to assess the impact of bias correction to RCM rainfall simulations from four different GCMs. The study was conducted in Victoria Australia and the hydrologic impact of the bias correction was assessed by looking at simulated runoff from 10 catchments. Quantifying the impact of bias correction on climate projection analyses is definitely an important research topic.

Main Comments: This is a regional study that is carried in Victoria Australia and therefore it is likely dependent on the modeling framework and the region's characteristics.

[Figure]

Thus, beyond the overall educational value that the readers who are not familiar with the region may gain, in order to appreciate the results and understand their applicability to other regions, the authors should describe in much more details the relevant regional hydrological characteristics, some differences among the selected catchments, the region's climate, and review the projected climatic change. The current version provides very little information on the study region.

The authors use QQM procedure to bias correct the WRF rainfall [BC-WRF]. A description of the procedure is missing and even the seasons that were used for the correction are not specified. The authors claim in a few places that the BC time series underestimate the daily wet-wet transition. Using the QQM procedure should only correct the magnitude of the [daily] rainfall events. Therefore, the dry-wet transitions of the rainfall from the raw WRF should not be different in the bias corrected rainfall. In addition, the spatial correlation of the BC-WRF, which is found to be different than the observed, is also should not be different than the raw WRF. These uncertainties in the rainfall sequencing and spatial correlation are likely originated from the GCMs and RCM and not from the bias correction procedure.

The four GCMs that were selected for this study should be validated with respect to their historic rainfall simulations . Their representation of the regional climatology in time and space should be evaluated. The raw GCMs should also be compared to analyze their projected climate change signal. The VERY large biases (hundreds of mm) of the raw-WRF (Fig 2) raise suspicion that the model may not capture the climatological features of the region. As for the GCMs evaluation, the raw-WRF simulations also have to be assessed in time and space to verify that it gets the seasonality and the expected spatial distribution. The authors may decide to remove the simulations GCMs and raw-WRF simulations that do not capture basic climatological features.

In Figure 8 the percent of the projected change in the runoff is presented. This analysis could be augmented by showing the transition of the percent change from the GCM, raw-WRF, BC-WRF, raw runoff and BC runoff. In the current analysis, the differences

in changes are mainly attributed to the selection of the GCM, and the contribution of the BC is unclear

The hydrologic model that is used in this study (equations are not given) has 4-parameters. However, only one parameter (x1) the soil moisture storage represents the rainfall-landsurface interaction. The three other parameters control the routing. Therefore, in annual time scale and from mass balance perspective the most sensitive parameter should be x1. The use of such a simple hydrologic model can be an advantage because it is possible to conduct a sensitivity analysis to assess the dependency of the BC on the soil parameters. Thus, the contribution of the model to the impact of BC can be assessed.

A recent relevant publication that evaluates the uncertainty in WRF dynamic downscaling to water resources application:

Shamir E., E. Halper, T. Modrick, K. P. Georgakakos, H-I. Chang, T. M. Lahmer, C. Castro. 2019. Statistical and dynamical downscaling impact on projected hydrologic assessment in arid environment: A case study from Bill Williams River basin and Alamo Lake, Arizona": https://www.sciencedirect.com/science/article/pii/S2589915519300033

Minor comments The title is misleading: I recommend to revise the title to: 'Impact of bias corrected downscaled rainfall on projected future runoff'. The current title assumes that there are biases and it is not clear if the biases stem from the GCMs or RCM.

Unit of grid cells should be either 10x10 km or 10 km2, not 10 km. This should be fixed throughout the manuscript

'underestimation biases in wet-wet transition probabilities' See my comment above that probability matching does not correct for transition. In addition, the underestimation is not of the 'probabilities'. Maybe you meant to say that it underestimate the wet-wet transition occurrences.

Describe the emission scenario that was selected for this study 'SRES A2' is insufficient description.

The selection of 2060-2079 as the period for analysis of projected change is untraditional. Eqn 1 presents an uncommonly used objective function. The authors should discuss the reasons to select it and what hydrologic features this function emphasizes

The statement 'The lumped modelling generally produced a slightly better calibration than the distributed modelling (Andréassian et al., 2004)' seems like a general statement. It will be interesting to state the results of your comparison between the two approaches.
* * *

---

## Author Response (AR1)

We thank the Editor for requesting we re-submit a revised manuscript addressing the Reviewer's comments and suggestions. Please see below the details of our modifications in the revised manuscript.

**Short Comment 1**

1. [D]o you think the biases in runoff projection are also limited to the hydrological model choice? More appropriately, can the biases be reduced by employing a semi or fully-distributed hydrological models which account for land use, soil characteristics and sub-surface flows? This is because there are several existing studies which have validated that runoff predictions are not only limited to climatological data, but physical processes such as SW-GW interaction, vegetation cover etc. which GR4J does not count for.

   *It is recognized that the choice of hydrological model should depend on the purpose of the model application. In this research, since our purpose is focused on investigating the bias in the projected total runoff without accounting for other drivers of hydrological change (land use, etc), we believe that the choice of a conceptual lumped rainfall-runoff model is efficient and effective given that the model has been well-calibrated to reproduce the observed runoff in the baseline period. We accept that a semi of fully distributed hydrological model could account for land use, soil characteristic and SW-GW interaction. However, there is no guarantee that the semi or fully distributed hydrological model would outperform the lumped model in reproducing the runoff of the baseline period. This is simply because that the semi or fully distributed model, although having more precise descriptions of the hydrological processes, has more parameters to be calibrated and more processes to be validated. If there isn't sufficient information available and the semi or fully distributed model is calibrated and validated against the observed streamflow (as that for the lumped model), the difference of the performance between the lumped model and the semi/fully distributed model could be marginal compared to the bias caused by the climate input errors. Nevertheless, it would be an interesting topic to explore and quantify the uncertainties in runoff projection in the future from using different complexities of hydrological models.*

2. Based on your experience do you think if the bias correction (while using either QQM or linear scaling) of the GCM projections was done at station scale and used for the catchment scale (lumped) hydrological simulation would have reduced the bias in the results compared to the results presented (as wide range of results are observed for the four GCM projections) in your manuscript?

   *Firstly, just to clarify, we use RCM projections that are boundary-forced by the GCMs and not the GCM projections themselves. The bias characteristics of the gridded (10 x 10 km) RCM rainfall would still be present if bias-correcting to station data. Thus, the same wide range of results would occur for the catchment scale (lumped) hydrological simulation whether using station or gridded bias corrected data inputs.*

3. Pg4 Ln 16: Can you please elaborate on the use of GCM data for only SRES A2 scenario? Also, as the SRES emission scenario data inherit more uncertainties over the RCP scenarios, is it valid for its use in this study as this paper deals with uncertainties cascading to hydrological simulations? This is because, in a study such as Woldemeskel et al. (2015), the uncertainty in precipitation for RCP scenarios of CMIP5 were significantly lower compared to SRES scenarios of CMIP3 GCMs in the Australian region. What do you think about this?

*The suite of WRF runs we have access to for this study only have CMIP3 GCMs forcing for the A2 SRES scenario. In future work we will assess, when available, CMIP5 RCP-based WRF runs. Referring to Woldemeskel et al. (2016), their Figure 4 of shows that CMIP3 and CMIP5 precipitation projections results for Southern Australia are similar.*

*We have amended the manuscript and added the text:*

*Whilst the SRES A2 is from the previous generation of Coupled Model Intercomparison Project CMIP3 scenarios (Nakicenovic N et al., 2000), it is relevant to assessing plausible climate change impacts as Woldemeskel et al. (2016) have shown that CMIP3 and CMIP5 projected precipitation changes are similar for Southern Australia.*

4. Can you please write more on the calibration of GR4J at grid scale? As per my experience, GR4J accounts for the rainfall and PET to determine the effective precipitation and then the flow routing is done at catchment outlet. As per the description given, it is not clear how the model is set using the distributed method at 10 km grid size? My specific question is how the routing is done at 100 km2 grid? Also, how the calibration is done at 100 km2 grid?

*The underlying rationale for using the distributed simulation is to investigate the impacts of spatial correlation in climate variables on runoff projection. For this purpose, in the distributed method, we assume that each 10 x 10 km grid cell in the catchment shares the same GR4J parameter set but has different climate inputs. Given a specific parameter set, the hydrological model runs at each 10 x 10 km grid cell (sub-catchment) and the simulated streamflow for each grid cell is summed up according to their area-weight to represent the total streamflow at the outlet of the catchment, which is then compared against the streamflow observed at the outlet gauge. The parameters are then calibrated to minimize the objective function described at Eq.1 based on the area-weighted sum of the simulation streamflow and the observed streamflow at the outlet of the catchment. This is described in the manuscript on page 6 under the section '2.2 Hydrological model'. When describing the whole of Victoria runoff simulation, we have added to the description the additional text:*

*There is no routing between grid cells in this case.*

5. Pg2Ln 20 and 21: Please be consistent with the use of "-" in BC WRF as in some places it is there and some places not. Same issue with the "BC-rainfall" and "BC rainfall". Please check.

*We have made the terminology consistent throughout.*

6. Pg3 Ln6: Typo "WFR" instead of "WRF".

*We have corrected.*

7. Pg3Ln 3-7: I understand there are several advantages of using the AWAP data, however, as per the study of Tozar et al. (2012), using gridded data (be it either AWAP or SILO) always introduce artificiality (due to spatial and temporal interpolation) and alter the "realness". This further cascades to hydrological simulations and leads to unrealistic results. Do you think using AWAP dataset is valid rather than using BoM gauges data for this study?

*We agree that using gridded data, with its inherent interpolation errors, does alter observed properties in comparison to station 'point' data. However, we think it is valid to use the AWAP dataset in our study because our 'observed' runoff is that simulated by GR4J when driven with the AWAP data. Given this is the baseline to which we compare WRF driven results, the use of AWAP is not a source of bias for our study.*

8. Can you please briefly write about the quantile-quantile mapping approach used in the study? It will be useful for the readers from non-climatological background.

*We do not want to repeat too much from the companion paper by Potter et al. (2019) (https://doi.org/10.5194/hess-2019-139) as that paper contains the full details of the quantile-quantile mapping approach used.*

*We have added the text:*

*The QQM-BC approach, using the R-package 'qmap' using the methodologies developed by Gudmundsson et al. (2012), is applied for each three-month season (i.e. DJF, MAM, JJA and SON) to each grid cell independently, mapping the simulated (WRF) to the observed (AWAP) daily rainfall cumulative density function quantiles such that a WRF simulated amount is replaced with the observed rainfall amount for the corresponding percentile, with linear interpolation between percentiles and for upper tail simulated amounts greater than observed.*

9. Pg5 Ln4: Can you please provide a list of all these 137 catchments in the supporting information? Can you also please provide the median and the percentile of the GR4J calibration parameters for both catchment scale and grid based calibration in the supporting information file?

*We have produced supporting information outlining the distributions of the parameters of GR4J calibration for the 137 catchments. As we outline in our description of the methods used, the lumped calibration is conducted only for the catchments listed at Table 1. For all the grids of the Victorian domain, GR4J is parameterized based on the parameter sets from the 137 catchments using the nearest neighbour approach.*

10. Pg5 Ln 23-24: Can you briefly mention the probable reasons for the better performance of the lumped approach over the distributed approach of GR4J calibration? Given that the catchments are flat (as majority of catchments in the SE Australian region are flat), a distributed approach should have yielded in better results.

*We have added the following text to the manuscript:*

*This is because the underestimation in spatial correlation in daily rainfall amounts between the individual grid points of the WRF rainfall simulations, compared to AWAP, results in distributed GR4J calibration underestimating runoff. The lumped rainfall compensates for the spatial underestimation and hence runoff simulations using lumped BC WRF rainfall are closer to those obtained using lumped AWAP rainfall.*

11. Are the runoff simulation results presented in sections 3.1 and 3.2 based on the distributed GR4J model setup?

*Yes, for the Victoria state-wide results in these sections the results are for each grid cell, i.e. the distributed GR4J model setup.*

12. For the future plans (as mentioned in the conclusion), I would also recommend the use of a physical-process based semi-distributed (if not fully distributed) hydrological model for the future hydrological predictions.

*Thanks for the recommendation. We agree that it is worth exploring the potential of the application of semi- or fully-distributed hydrological models in projecting hydrological*

*changes under future climate. The application of such distributed models could be expected to provide more physically interpretable modelling at the catchment scale, but their implementation is challenging. For example, they are more likely to have equifinality problems in calibrating the distributed hydrological model given limited measurements and more parameters and if the parameters are assumed to vary spatially. The equifinality could therefore lead to difficulties in transferring parameters from the gauged catchments to the ungauged catchments if we aim to provide the projections at the regional scale across the entire Victorian domain.*

**Referee comment 1**

1.  An important part of your work analyses empirically scaled observed data. It might be useful for the readers if you describe a bit of the method from Chiew et al. (2009).

    *We have added the description:*

    *This empirical scaling method (Chiew et al., 2009) rescales the historical grid-cell timeseries by multiplicatively applying the changes between historical and future period climate model projections. This can be applied on an annual basis, on a seasonal basis, or in a two-step process first on a seasonal and then on an annual to maintain the overall annual change as shown by the climate model.*

2.  Similar to the above, you could provide a few sentences to describe the QQM bias correction method.

    *We do not want to repeat too much from the companion paper by Potter et al. (2019) ([https://doi.org/10.5194/hess-2019-139](https://doi.org/10.5194/hess-2019-139)) as that paper contains the full details of the quantile-quantile mapping approach used. We have added the text:*

    *The QQM-BC approach, using the R-package 'qmap' using the methodologies developed by Gudmundsson et al. (2012), is applied for each three-month season (i.e. DJF, MAM, JJA and SON) to each grid cell independently, mapping the simulated (WRF) to the observed (AWAP) daily rainfall cumulative density function quantiles such that a WRF simulated amount is replaced with the observed rainfall amount for the corresponding percentile, with linear interpolation between percentiles and for upper tail simulated amounts greater than observed..*

3.  In future work, such as the one stated at the end of the conclusions, I would recommend using the last generation of climate models that is available as this would provide an analysis of the current state-of-the-art.

    *We agree and will clarify in the manuscript that the WRF simulations forced by CMIP3 GCMs were the only ones available to us at the time. Newer CMIP5 GCM forced continuous (rather than time-slice) WRF runs are becoming available and we will use these in future research.*

4.  I understand that this works builds up from previous studies, however, could you give a sentence on why you (or the studies that your work builds on) choose to use the WRF model? Would you expect similar results from using other RCMs? The above applies for the hydrological model, would you expect that a fully-distributed integrated model will have different result?

    *At the time of this study, we only had access to WRF runs at the scale of 10 x 10 km so WRF results were chosen as an 'ensemble of opportunity'. Subsequently, an additional RCM (CCAM; see [https://climatechangeinaustralia.gov.au/en/climate-projections/future-climate/victorian-climate-projections-2019/](https://climatechangeinaustralia.gov.au/en/climate-projections/future-climate/victorian-climate-projections-2019/) ) has produced 5 x 5 km output over Victoria forced by 6 CMIP5 GCMs for RCP4.5 and RCP8.5 with continuous daily data available to 2100. We are currently assessing these new data with a view to use them in future research. So far, we are finding that CCAM projections are drier than WRF. We have added the statement:*

    *Also, other RCMs could produce different results when forced by the same GCMs.*

*In terms of runoff projection at catchment scale, it can be expected that the difference between the results of a fully-distributed model and that of our distributed modelling (lumped model but used in a distributed way) would be marginal if both models have been well calibrated and are of similar bias. This is because that both modelling approaches (i.e., fully-distributed model and our distributed modelling) have consider the effects of the spatial variation (or spatial cross-correlation) of climate variables, which is the dominant driver affecting runoff. However, in contrast to the distributed modelling approach described in our manuscript, we recognize that the parameters of a fully-distributed model are not uniform across the catchment, which implies that the hydrological responses could vary spatially across the catchment. Hence, if the fully-distributed model is validated to be capable of reflecting the spatial heterogeneity of hydrological response and the projected changes in climate are of highly spatial variation, theoretically, the projected runoff change could be different. For example, assuming a dominant sub-catchment experiences much higher precipitation reduction than other sub-catchments, we can expect bigger difference between the fully-distributed model and the distributed modelling described in our manuscript. To reflect this difference appropriately, however, more information is needed to validate the model and ensure the fully-distributed model performs realistically across the catchment to avoid the possible mis-matching of hydrological responsive area and the climate sensitive area. This is especially important for large-scale catchments with higher spatial heterogeneity. As the catchments in this studied are all small-scale (around 1000km²), the hydrological effects of spatial heterogeneity are therefore assumed to be negligible as compared to the relatively large bias in the projected climate variables.*

5. Pg.1 L27 - The comma is missing in the following: Thus, 'bias correction' methods.

   *Corrected.*

6. In Pg. 2 Lines 2 to 9 - You are referring to some of the limitations of bias correction. I think it would be also important to include the stationarity assumption of the relationship between simulations and observations. I think this is the main concept that supports the idea of bias correction.

   *We agree that a fundamental assumption of bias correcting current and future periods is that the relationship derived for the current period is stationary and hence can be applied in the future period. We have added the text:*

   *Bias correction also assumes stationarity in the QQM relationship, assuming that the mapping derived for the historical period applies under future climate conditions (Teutschbein and Seibert, 2013).*

7. Pg.4 Lines 16 to 20 - It is not clear to me whether you use the monthly mean potential evapotranspiration or daily values. Also, you could include a sentence stating whether if including the potential evapotranspiration simulations could change your results considerably.

   *We use historical monthly mean values of daily potential evapotranspiration. Apologies this is not clear, we have clarified this in the revised manuscript, and added the text:*

   *Changes to PET would cause an additional reduction in runoff under a warming climate but would not change the relative results, as presented, in any considerable manner because the range of change is dominated by the large range in rainfall projections, i.e. PET is a 2nd order*

*effect. For example, for a similar region Potter and Chiew (2011) found increased PET only explained 5% of runoff reduction in a prolonged drought.*

*Potter, N.J. and Chiew, F.H.S. (2011) An investigation into changes in climate characteristics causing the recent very low runoff in the southern Murray-Darling Basin using rainfall-runoff models. Water Resources Research 47(12).*

8. Pg. 6 Line 16 – You could say how large is the bias of the 99th percentile in rainfall as it can give a background for the bias in the 99th percentile of runoff.

   *We have added the text:*

   *The residual bias in the 99th percentile daily rainfall error has a mean of 0.02, 0.01, 0.02, 0.06, 0.04 and 0.03 mm for the WRF runs forced by NCAP/NCAR Reanalysis, ERA-Interim Reanalysis and the historical runs from the CCCMA3.1, CSIRO-MK3.0, ECHAM5 and MIROC3.2 GCMs, respectively. These amounts correspond to percentage errors of 0.06, 0.01, 0.10, 0.24, 0.14 and 0.11% for these runs, respectively. These are mean errors across all grid-cells, with the error for individual grid-cells ranging from -2.1 mm to +3.0 mm, or 7.5% to +10.1%.*

9. Pg 9. Lines 17 to t20 – Can you include some reference that assess whether the model is good to simulate runoff on a changing catchment. I am not sure that there is a reference for the GR4J model. If not, you can acknowledge that there is no previous study analyzing it.

   *There are numerous publications on using hydrological models (lumped, semi-distributed or fully-distributed) to assess the hydrological impacts of climate. The GR4J is just one of the lumped hydrological models widely used and testified to be competitive among currently available hydrological models. The GR4J model is capable to provide hydrological projection informed by climate change. However, this does not necessarily mean the GR4J can be used to simulate runoff under all possible catchment changes beyond climate (like land use and land cover change, human impacts) as the model itself is conceptual and not physically represents all the hydrological processes. We have added the text:*

   *We note that GR4J model performance for future climate conditions differing from the calibration period are an additional source of uncertainty (Stephens et al., 2019), but we do not assess such potential deficiencies here.*

   *Stephens, C. M., Marshall, L. A., and Johnson, F. M.: Investigating strategies to improve hydrologic model performance in a changing climate, Journal of Hydrology, 579, 124219, https://doi.org/10.1016/j.jhydrol.2019.124219, 2019.*

10. Pg. 11 Lines 1 and 2 – You could also include the opposite view that bias-corrected outputs should be used with care giving that they are not based in any knowledge of the clime at physics (Ehret et al 2012).

    *We agree that there are such concerns and will cite Ehret et al (2012) to highlight that this is an issue. Our text now reads:*

    *There is an argument that, because the QQM-BC rainfall corrects distributional biases in the raw WRF rainfall, the QQM-BC rainfall change signals are more realistic than those of the original WRF-raw rainfall, however (Ehret et al., 2012) are cautious of this view stating QQM-BC does not have physical justification.*

11. Pg. 11 last line and page 12 first line – Correct the sentence as it does not make sense at the moment.

    *There is a comma missing as this should be a continuation from the previous sentence, we have revised to read: "The differences in runoff changes from empirically scaled rainfall (i.e. based on seasonal and/or annual changes in WRF-raw rainfall) and BC rainfall are shown in Figure 5 to Figure 7 for annual mean runoff, daily 99th percentile flow and the number of days above observed 95th percentile flow, respectively."*

12. Table 1. Given that the high flows are relevant for this research, consider including here how the models do when simulating the high flows (i.e. 99th percentile). This will increase the credibility of the results (even if you use the simulated, and not the observed, runoff as benchmark for comparison.)

    *We do not understand this request given Table 1 presents mean simulated runoff as attributes, not as validation statistics. So, we're not sure how showing 99$^{th}$ percentile flows would "increase the credibility of the results"?*

13. Figure 1. You can add the legend title (i.e. annual mean rainfall (mm/yr)"

    *Figure caption includes the legend title.*

14. Figure 9. Add the title of the y axis title.

    *Do you mean x-axis title, as the y-axis are all labelled? We have modified the figure caption to make it clear we are showing high flow percentiles for the 60 to 99 percentiles.*

**Referee comment 2**

1. This is a regional study that is carried in Victoria Australia and therefore it is likely dependent on the modeling framework and the region's characteristics. Thus, beyond the overall educational value that the readers who are not familiar with the region may gain, in order to appreciate the results and understand their applicability to other regions, the authors should describe in much more details the relevant regional hydrological characteristics, some differences among the selected catchments, the region's climate, and review the projected climatic change. The current version provides very little information on the study region.

   *We will add more detail, as requested, regarding the regional and catchment hydrological characteristics and a review of projected climate change. Rather than add a lot of detailed descriptions, we have summarised the climatology (temperate; wet mild winters, dry hot summers) and runoff. Our Figure 1 shows the range of mean rainfall for the catchments. We also cite other recent research detailing catchment characteristics:*

   *Chiew, F. H. S., Potter, N. J., Vaze, J., Petheram, C., Zhang, L., Teng, J., and Post, D. A.: Observed hydrologic non-stationarity in far south-eastern Australia: implications for modelling and prediction, Stochastic Environmental Research and Risk Assessment, 28, 3-15, 10.1007/s00477-013-0755-5, 2014.*

   *Chiew, F., Zheng, H., and Potter, N.: Rainfall-Runoff Modelling Considerations to Predict Streamflow Characteristics in Ungauged Catchments and under Climate Change, Water, 10, 1319, 2018.*

   *Regarding projected climate change, we will add a citation of a recent detailed summary of the projected change for the region – Victoria's Climate Science Report 2019 (*https://www.climatechange.vic.gov.au/__data/assets/pdf_file/0029/442964/Victorias-Climate-Science-Report-2019.pdf*) The Executive Summary of this report refers to a high degree of consensus in projected rainfall declines: "Annual rainfall is projected to decrease across the state, due to declines across autumn, winter and spring. When extreme rainfall events do occur, they are likely to be more intense. Areas of the Victorian Alps are projected to see a greater reduction in rainfall than the surrounding areas."*

2. The authors use QQM procedure to bias correct the WRF rainfall [BC-WRF]. A description of the procedure is missing and even the seasons that were used for the correction are not specified.

   *We do not want to repeat too much from the companion paper by Potter et al. (2019) (*https://doi.org/10.5194/hess-2019-139*) as that paper contains the full details of the quantile-quantile mapping approach used. We have added the text:*

   *The QQM-BC approach, using the R-package 'qmap' using the methodologies developed by Gudmundsson et al. (2012), is applied for each three-month season (i.e. DJF, MAM, JJA and SON) to each grid cell independently, mapping the simulated (WRF) to the observed (AWAP) daily rainfall cumulative density function quantiles such that a WRF simulated amount is replaced with the observed rainfall amount for the corresponding percentile, with linear interpolation between percentiles and for upper tail simulated amounts greater than observed.*

3. The authors claim in a few places that the BC time series underestimate the daily wet-wet transition. Using the QQM procedure should only correct the magnitude of the [daily] rainfall events. Therefore, the dry-wet transitions of the rainfall from the raw WRF should not be different in the bias corrected rainfall. In addition, the spatial correlation of the BC-WRF, which is found to be different than the observed, is also should not be different than the raw WRF. These uncertainties in the rainfall sequencing and spatial correlation are likely originated from the GCMs and RCM and not from the bias correction procedure.

*We agree. We believe this is covered in our introduction, where we state:*

*" Potter et al. (2019) found that the quantile mapping (QQM) bias correction approach used to correct raw WRF daily rainfall, applied on a cell by cell seasonal basis, does not correct for underestimation biases in wet-wet transition probabilities. Hence, we show, BC WRF rainfall will tend to underestimate runoff compared to runoff simulated using observed rainfall. At the catchment scale both raw and BC WRF rainfall underestimate spatial correlation between cells within a catchment, which is an additional source of runoff uncertainty."*

*It is implicit in this statement that the errors/biases in rainfall sequencing and spatial correlation originate from the GCMs and RCM and not from the bias correction procedure.*

4. The four GCMs that were selected for this study should be validated with respect to their historic rainfall simulations. Their representation of the regional climatology in time and space should be evaluated. The raw GCMs should also be compared to analyze their projected climate change signal. The VERY large biases (hundreds of mm) of the raw-WRF (Fig 2) raise suspicion that the model may not capture the climatological features of the region. As for the GCMs evaluation, the raw-WRF simulations also have to be assessed in time and space to verify that it gets the seasonality and the expected spatial distribution. The authors may decide to remove the simulations GCMs and raw-WRF simulations that do not capture basic climatological features.

*We refer to Evans et al. (2014) regarding the selection of the GCMs (Page 4, line 4). We have included more detail on the rationale used in GCM selection, adding the text:*

*Evans et al. (2014) described the experimental design for these WRF downscaled simulations, outlining the selection of four CMIP3 GCMs (CCCM3.1, CSIRO-Mk3.0, ECHAM5 and MIROC3.2-medres) based on a three-stage process: (1) the performance of a total of 23 CMIP3 GCMs was evaluated, and models that did not adequately simulate the historical climate of south-eastern Australia were rejected; (2) the set of GCMs that performed well was then ranked on the basis of a measure of their independence; and (3) the GCMs were then evaluated on the basis of their projections of future climate change. The four most independent models that spanned the largest range of plausible future climates were chosen.*

*Regarding the Reviewer's suspicion that WRF may not capture the climatological features of the region, we refer to our WRF assessment description on page 5, including Olson et al. (2016) who did report biases in rainfall climatology.*

5. In Figure 8 the percent of the projected change in the runoff is presented. This analysis could be augmented by showing the transition of the percent change from the GCM, raw-WRF, BC-WRF, raw runoff and BC runoff. In the current analysis, the differences in changes are mainly attributed to the selection of the GCM, and the contribution of the BC is unclear.

*Referring to rainfall, we do not use GCM rainfall and so cannot show "the percent change from the GCM"; we do show the raw and BC WRF rainfall change in Figure 4. For runoff, because of the biases in raw WRF rainfall, it does not make sense to show the runoff change that results from using raw WRF rainfall.*

6. The hydrologic model that is used in this study (equations are not given) has 4-parameters. However, only one parameter (x1) the soil moisture storage represents the rainfall-landsurface interaction. The three other parameters control the routing. Therefore, in annual time scale and from mass balance perspective the most sensitive parameter should be x1. The use of such a simple hydrologic model can be an advantage because it is possible to conduct a sensitivity analysis to assess the dependency of the BC on the soil parameters. Thus, the contribution of the model to the impact of BC can be assessed.

   *Thanks for the comments and the interesting suggestion to quantify the contribution of the hydrological model to the impact of BC. However, we'd like to clarify that the purpose of this study is to assess the impact of BC on the projected runoff. To do the projection, we first calibrated the hydrological model using the observed climate (not from the GCM or RCM, so no BC involved) and fixed the calibrated parameters for the model to project possible runoff change. Although it would be interesting to assess the sensitivity of projected runoff on BC rainfall and hydrological parameters (e.g., conduct sensitivity analysis by varying the soil parameter at specified small intervals), to compare the contribution from BC and parameter uncertainties, this is not within the scope of our study.*

7. The title is misleading: I recommend to revise the title to: 'Impact of bias corrected downscaled rainfall on projected future runoff'. The current title assumes that there are biases and it is not clear if the biases stem from the GCMs or RCM.

   *The paper looks at the change in downscaled rainfall as applied through empirical scaling, as well as the bias corrected downscaled rainfall. We therefore believe our original title is more encompassing of the work as presented. However, we are happy to follow the Editor's guidance as to whether a title change is required.*

8. Unit of grid cells should be either 10x10 km or 10 km2, not 10 km. This should be fixed throughout the manuscript.

   *Yes the grid cells have a 10 km length scale, and we have reported as 10 x 10 km throughout in the revised manuscript.*

9. 'underestimation biases in wet-wet transition probabilities' See my comment above that probability matching does not correct for transition. In addition, the underestimation is not of the 'probabilities'. Maybe you meant to say that it underestimate the wet-wet transition occurrences.

   *The term 'transition probabilities' is commonly used in the literature when referring to the transition properties of wet- and dry-day sequences. They are true probabilities, bounded to be from 0 to 1.*

10. Describe the emission scenario that was selected for this study 'SRES A2' is insufficient description.

    *We have added the description:*

*"The SRES A2 scenario describes a very heterogeneous world with high population growth and technological change that is fragmented and slow (Nakicenovic et al. 2000). The SRES A2 emission scenario was selected for the NARCliM climate projections because the global emissions trajectory suggested that it was the most likely scenario. Recent publications have confirmed that we are tracking at the higher end of the A2 scenario (Peters et al. 2013)."*

*Nakicenovic N , Alcamo J, Grubler A , Riahi K , Roehrl RA, Rogner H-H, & Victor N. Special Report on Emissions Scenarios (SRES), A Special Report of Working Group III of the Intergovernmental Panel on Climate Change. Cambridge: Cambridge University Press, 2000.*

*Peters GP, Andrew RM, Boden T, Canadell JG, Ciais P, Le Quere C, Marland G, Raupach MR, Wilson C. Commentary: The challenge to keep global warming below 2 degrees C Nature Climate Change, 3:4-6, 2013.*

11. The selection of 2060-2079 as the period for analysis of projected change is untraditional.

    *The periods for which WRF runs were undertaken was set by the research group running the WRF model for south-eastern Australia and we had no input to the choice of periods. The periods selected by them were 1990 to 2009 (base), 2020 to 2039 (near future), and 2060 to 2079 (far future).*

12. Eqn 1 presents an uncommonly used objective function. The authors should discuss the reasons to select it and what hydrologic features this function emphasizes.

    *The Eqn1 is an objective function combing the commonly used NSE and the Bias. Different to the conventional NSE, the objective function puts more weight on the Bias term. The purpose of this combined objective function is to ensure that the bias of the model can be constrained to an acceptable level (as noted in the reference we provide, Viney et al. 2009) as we are targeting the projection of total runoff. We have added the text:*

    *"This objective function, from Viney et al. (2009), combines the commonly used NSE and the Bias to constrain total model bias in runoff simulation."*

13. The statement 'The lumped modelling generally produced a slightly better calibration than the distributed modelling (Andréassian et al., 2004)' seems like a general statement. It will be interesting to state the results of your comparison between the two approaches.

    *This statement refers to the specific results as shown in Table 1, which shows that for 9 of the 10 catchments investigated the 'lumped' distribution has a higher NSE than the distributed case. We have joined this sentence with the prior sentence to make the link to Table 1 explicit.*

[revised manuscript text omitted]

---

## Author Response (AR2)

We have made all the corrections to figures as requested below.
Technical corrections
Figure 1 - please add coordinates and/or scale bar. In addition, please add a caption to the legend.

Figure 2 - here also (and in the other figures), please add a caption to the legend.

Figure 8 - please label the axes and add a legend to the symbol colours.

Figure 9 - text is too small, can you increase the font size? same for figure 10.